# Recognising Religious Groups as Litigants: An International Law Perspective

**Mark Fowler** [1,*] **and Alex Deagon** [2]

1    School of Law, University of New England, Armidale, NSW 2350, Australia
2    School of Law, Queensland University of Technology, Brisbane City, QLD 4000, Australia; alex.deagon@qut.edu.au
*    Correspondence: mfowler@fclaw.net.au

**Abstract:** The Australian Human Rights Commission has claimed that recognising religious corporations as litigants in religious discrimination claims departs from international human rights law, which only protects the rights of natural legal persons. In this article we respond to that claim by arguing that under international law, Australia should protect the ability of religious groups to be litigants, including corporations. The International Covenant on Civil and Political Rights requires Australia to respect and ensure individuals have the right to manifest their beliefs in community with others, and that such communities are protected against discrimination on religious grounds. This requirement entails granting religious groups the ability to pursue legal measures to preserve the enjoyment of these rights by their members.

**Keywords:** human rights; religious freedom; international law; constitutional corporations; anti-discrimination law

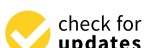



## 1. Introduction and Scope

In the wake of religious freedom concerns stemming from Australia legally recognising same-sex marriage in 2017, the-then Prime Minister Malcolm Turnbull instituted a panel to evaluate protection for the human right to freedom of religion and belief in Australia. This panel, chaired by former conservative Government minister Philip Ruddock, became known as the 'Ruddock Review'. The Ruddock Review ultimately concluded that legal protection for religious freedom in Australia is inadequate and recommended that a bill be passed to protect against discrimination on the grounds of religion or belief. In 2019 the-then Attorney-General Christian Porter released an initial exposure draft of the bill that received criticism from religious groups, LGBT communities, and human rights groups. A second exposure draft of the bill was subsequently released that received greater support from religious groups, but many of the concerns of LGBT and human rights advocates remained unaddressed.[1] Debate on the bill remained largely dormant over 2020 and 2021 as the COVID-19 crisis took priority. Attorney-General Michaela Cash subsequently released a revised Religious Discrimination Bill (the RDB or Bill) in late 2021. This Bill was then endorsed by two Parliamentary Inquiries and subsequently passed by the lower House of Parliament, only to then be withdrawn (by the Government) from consideration in the Senate. According to the-then Prime Minister Scott Morrison, the reason for this withdrawal was the possibility that separate amendments to the *Sex Discrimination Act 1984*, which had passed the House of Representatives on the same evening as the Bill's passage, would also pass the Senate. At the subsequent Federal election the Morrison Government was defeated and the Bill lapsed. The victorious Labor Party has committed to introduce legislation that will prevent discrimination against people of faith. At the time of writing, however, the substantive legislative provisions that will give effect to this proposal are not known.

---

1    For a general overview see (Sarre 2020).

The terms of reference of a referral to the Australian Law Reform Commission concerning religious educational institutions included a November 2022 request for the Commission to consider if some or all of the reforms recommended as a result of the inquiry could be included in a Religious Discrimination Bill.

One consideration that arises is if non-natural legal persons (i.e., corporations and unincorporated groups) should receive the benefit of any legislative protection. This consideration is central to religious discrimination protections because of the universally recognised propensity of religious believers to engage in mutual undertakings through institutional forms,[2] as acknowledged by Appeal Judge Redlich, who put it mildly when he said '[c]orporations have a long history of association with religious activity'.[3] If religious institutions are precluded from religious discrimination protections, these protections will fail to adequately protect religious believers from discrimination. During the course of consultation on the original Bill, the question of if religious institutions could rely on the protections was the subject of some disputation. For example, in their submission on the Religious Discrimination Bill the Australian Human Rights Commission argued that the first exposure draft of the Bill was too broad in defining who may be a victim of religious discrimination, asserting that the ability of religious corporations, such as religious institutions, schools, charities and businesses, to make discrimination claims would be a significant departure from international human rights law, which only protects the rights of natural persons.[4] This article responds to that claim by arguing that, in order to adequately give effect to the protections afforded under international law, Australia should protect the ability of religious corporations to be complainants.

Part II outlines the general obligations under the *International Covenant on Civil and Political Rights*, whose Articles 2, 18 and 26, *inter alia*, require Australia to respect and ensure to individuals the right to manifest their beliefs in community (including through incorporated and unincorporated communities) with others, and to be protected against discrimination on religious grounds. In this respect, Part III considers various judicial pronouncements within domestic and international law that have addressed the question of if religious associations can sue for discrimination in their own right, arguing that existing Australian law gives rise to a lacuna in protections. Part IV applies these principles to argue that, as a result of Australia's international law obligations, Australia should recognise religious groups as litigants through religious discrimination legislation. Part V makes a brief comment on constitutional considerations before Part VI concludes.

Although the precise content of any potential Commonwealth protection of religious belief and activity is not known at the time of writing, discussion upon the provisions of the *Religious Discrimination Bill 2021* (Cth) pertinent to the protection of groups are addressed at particular points in order to develop the argument with sufficient particularity and to illustrate the application of the relevant principles of international law. While the focus of this article is primarily on the proposal for a Commonwealth law protecting groups against religious discrimination, the question of if religious groups may make discrimination claims demands attention across wide and multi-varied facets of Australian public life, and will accordingly be engaged. As will be further considered below, this is relevant to two separate current Federal Parliamentary Inquiries, the first being the Australian Parliamentary Joint Committee on Human Rights 'Inquiry into Australia's Human Rights Framework', focused on a proposed Australian Human Rights Act (AHRA), and the second being the Senate Legal and Constitutional Affairs Legislation Committee 'Inquiry into the Australian Capital Territory (Self-Government) Amendment Bill 2023',

---

2  See e.g., (Norton 2016; Harrison 2020). This article focuses on legal positions in domestic law and international human rights law, and not on conceptual distinctions, such as those between human rights inhering in natural persons (who may associate and act through group entities) versus human rights inhering in non-natural entities. For further discussion, see e.g., (Tasioulas 2012, pp. 17–61; David 2020).

3  See *Christian Youth Camps Ltd. v Cobaw Community Health Services Ltd.* 308 ALR 615 ('*Cobaw*'). [481] (Redlich J).

4  Australian Human Rights Commission, *Submission on the Religious Discrimination Bill*: Religious Freedom Bills | Australian Human Rights Commission.

which was prompted by the compulsory acquisition of Calvary Public Hospital by the Australian Capital Territory Government. The question is also relevant to State and Territory discrimination laws that each permit 'persons' to lodge a complaint of discrimination, rendering no distinction between natural and corporate persons. As a matter of statutory construction, the starting place for determining if 'person' includes a corporate person is the applicable interpretation Act, which each provide that reference to a 'person' includes a reference to a 'corporation'.[5] This presumption will apply in the absence of express or implied provisions to the contrary within the respective discrimination enactment.[6]

## 2. General Obligations under the ICCPR

The Religious Discrimination Bill purported to implement, among other instruments, Article 18 of the *International Covenant on Civil and Political Rights* ('ICCPR'), which was ratified by Australia in 1980. It states:

1.     Everyone shall have the right to freedom of thought, conscience and religion. This right shall include freedom to have or to adopt a religion or belief of his choice, and freedom, either individually or in community with others and in public or private, to manifest his religion or belief in worship, observance, practice and teaching.
2.     No one shall be subject to coercion which would impair his freedom to have or to adopt a religion or belief of his choice.
3.     Freedom to manifest one's religion or beliefs may be subject only to such limitations as are prescribed by law and are necessary to protect public safety, order, health or morals or the fundamental rights and freedoms of others.
4.     The States Parties to the present Covenant undertake to have respect for the liberty of parents and, when applicable, legal guardians to ensure the religious and moral education of their children in conformity with their own convictions.[7]

While freedom of religion, as recognised under the ICCPR, is a right exercised by individuals, Article 18(1) specifically protects the ability to act in community with others (including through unincorporated and incorporated vehicles) as a form of manifesting belief, both privately and publicly.[8] Article 18(4) obliges states to have respect for the liberty of parents to ensure the religious and moral education of their children in conformity with their own religious convictions, which in practice entails the creation and maintenance of associations formed for such purposes (faith-based schools are one example). In their 1993 General Comment 22 on Article 18, in paragraph 8, the Human Rights Committee has stated that the protection for parents in Article 18(4) 'cannot be restricted'[9] But, outside of this, the only permissible restrictions on religious manifestation are those found in Article 18(3), namely those that are *necessary* (not merely reasonable) to protect public safety, order, health, morals or the fundamental rights and freedoms of others. This is a high threshold that requires substantive proof before any legal limitation is appropriate.[10] In this respect

---

5     See, for e.g., *Acts Interpretation Act 1901* (Cth) s 2C; *Interpretation Act 1987* (NSW) s 21; *Acts Interpretation Act 1954* (Qld) 32D. To clarify this point, the Queensland Human Rights Commission has, for example, recently recommended that corporate bodies should be explicitly permitted to make a complaint of discrimination under the *Anti-Discrimination Act 1991* (Qld). See Queensland Human Rights Commission, *Building Belonging: Review of Queensland's Anti-Discrimination Act 1991* (2022): Recommendation 10.

6     For an example of how such statutory presumptions may be displaced, see the contrary view taken by Neave JA in *Cobaw* (n 3). This judgement related to an 'exemption' formerly provided to 'persons' under the *Equal Opportunity Act 2010* (Vic), and the expressed view applied to the 'exemption', as opposed to the separate question of if a religious body could be a complainant under the Act.

7     *International Covenant on Civil and Political Rights*, opened for signature 16 December 1966, 999 UNTS 171 (entered into force 23 March 1976).

8     See (Aroney 2014). As discussed below, other jurisdictions have recognised it can be exercised by corporations on behalf of individuals.

9     *Human Rights Committee, General Comment No 22: Article 18, 48th sess, (20 July 1993)*, [8].

10    In accordance with the Siracusa Principles, any restriction must be necessary to achieve one of the objects listed, and must be proportionate to that object, in the sense that it is the least restrictive means to achieve it: 'Siracusa Principles on the Limitation and Derogation of Provisions' in the *International Covenant on Civil and Political Rights Annex*, UN Doc E/CN.4/1984/4 (1984), https://www.uio.no/studier/emner/jus/humanrights/HUMR5503/h09/undervisningsmateriale/SiracusaPrinciples.pdf (accessed on 19 February 2019).

the right to manifest religious belief is distinct from the right to hold a belief under Article 18(1) which, like the parental right, is not subject to limitation.

Although Article 18 is self-standing, it also works in conjunction with the prohibition on discrimination found in Article 2.1 (which prohibits discrimination in the enjoyment of Article 18 rights, among other ICCPR rights) and Article 26 of the ICCPR (which is concerned more generally with equality before the law and equal protection of the law). They are, respectively:

> 2.1 Each State Party to the present Covenant undertakes to respect and to ensure to all individuals within its territory and subject to its jurisdiction the rights recognized in the present Covenant, without distinction of any kind, such as race, colour, sex, language, religion, political or other opinion, national or social origin, property, birth or other status.

> 26. Persons are equal before the law and are entitled without any discrimination to the equal protection of the law. In this respect, the law shall prohibit any discrimination and guarantee to all persons equal and effective protection against discrimination on any ground such as race, colour, sex, language, religion, political or other opinion, national or social origin, property, birth or other status.

In its General Comment on Article 26, the United Nations Human Rights Committee (UNHRC) states that:

> the term "discrimination" as used in the Covenant should be understood to imply any distinction, exclusion, restriction or preference which is based on any ground such as race, colour, sex, language, religion, political or other opinion, national or social origin, property, birth or other status, and which has the purpose or effect of nullifying or impairing the recognition, enjoyment or exercise by all persons, on an equal footing, of all rights and freedoms.[11]

There is an overlap between the prohibition of discrimination and the protection of religious freedom: where a domestic court denies a person's religious discrimination claim (assuming there are no other legitimate procedural, evidentiary or substantive grounds on which to deny the claim), it imposes an effective limitation on that person's religious manifestation, and the future manifestation of similar conduct in comparable circumstances. The overlap indicates that domestic religious discrimination protections can operate in a manner that fails to acquit the obligations imposed by Article 18(3). For example, indirect discrimination tests that permit limitations on religious manifestation when they are 'reasonable' admit a standard that may lead to the imposition of limitations that fail to acquit the 'necessary' limitations requirement imposed by Article 18(3).

Further, where a discriminatory act occurs against a corporation–for example, through the denial of services or venue booking, or the termination of a contract, such an act can amount to an effective limitation on the manifestation of the rights of persons associated with the corporation, including members of the corporation. Take a religious corporation that is refused a booking of a venue operated by government because the religious body intends to teach its traditional view of marriage during its use of the facility. As will be seen, under Australian discrimination law the proper litigant is the person that has been discriminated against, which in this case is the corporate body that sought the booking. The refusal has limited the manifestation of the rights of the speaker, and those who may wish to attend the proposed event. However, if the corporate body cannot be the subject of discriminatory action under domestic legislation it, and the religious believers connected to it, will not be protected. Given the unique propensity of religious belief to inspire community, and the accordant scope by which religious undertakings take place through legally recognised group structures, the ramifications are wide ranging, extending to the receipt of goods, services and facilities; the issuing of funding, grants or subsidies; and also licensing, accreditations and planning approvals. The immediacy of the issue has also

---

[11] See *Human Rights Committee, General Comment No 18: Non-discrimination, 37th sess, (10 November 1989)*, [7].

recently been highlighted by the Australian Capital Territory Government's compulsory acquisition of Calvary Public Hospital. The Territory Government is not relevantly bound by the provisions of local discrimination legislation. However, if it were so bound, the question would arise as to if the hospital, as a corporate body, could take the benefit of the protections where the acquisition could be evidenced to be 'on the ground of' the 'religious conviction' of the institution, or those of its associates.[12]

Having outlined the primary relevant international obligations, the following section develops the argument that Australia (inclusive of its States and Territories) should acquit its international obligations by enabling religious corporations as litigants in legislation that facilitates the group manifestation of religious belief by preventing discrimination against such groups. The same position would hold for the complainant enforcement mechanisms adopted under any proposed AHRA.

## 3. Domestic Law, International Law and the Rights of Religious Corporations

### 3.1. Corporations as Discrimination Complainants under Australian Law

In its submission on the Religious Discrimination Bill the Australian Human Rights Commission helpfully framed the argument against protecting corporate bodies from religious discrimination in the following succinct summary:

> it would be a significant departure from existing discrimination law, and from human rights law as protecting only humans, to allow corporations to bring actions for religious discrimination in their own right . . . The Commission recommends that the definition of 'person' be removed from the Bill and that the Explanatory Notes be amended to make clear that a complaint of discrimination on the ground of religious belief or activity may only be made by or on behalf of a natural person.[13]

The Commission asserts that the basis for this recommendation is that Commonwealth law does not recognise the ability of corporate bodies to receive protection from anti-discrimination law, and that international law does not protect the rights of religious bodies. For the reasons set out below, both assertions are subject to important qualification. As noted in the Introduction, most Australian discrimination law provides that a 'person' may initiate a complaint, with the default position that this would include corporate 'persons'. This was the view taken by Mason J in applying the *Racial Discrimination Act 1975* ('RDA') in *Koowarta v Bjelke-Petersen:*

> It is submitted that because, generally speaking, human rights are accorded to individuals, not to corporations, "person" should be confined to individuals. But, the object of the [International Convention on the Elimination of All Forms of Racial Discrimination] being to eliminate all forms of racial discrimination and the purpose of s. 12 being to prohibit acts involving racial discrimination, there is a strong reason for giving the word its statutory sense so that the section applies to discrimination against a corporation by reason of the race, colour or national or ethnic origin of any associate of that corporation. It is also submitted that the reference in the concluding words to "any relative or associate of that second person" is inappropriate to a corporation. Certainly that is so of "relative", but a corporation may have an "associate". The concluding words are therefore quite consistent with the "second person" denoting a corporation as well as an individual.[14]

The 'second person' referred to in the relevant provisions is the 'person', whether corporation or individual, who is the subject of a discriminatory act. Justice Mason thus clearly holds that, notwithstanding his recognition of the principle that 'human rights are

---

12  See *Discrimination Act 1991* (ACT) s 7(1).
13  See Australian Human Rights Commission (n 4) [58].
14  See (1982) 153 CLR 168, 236 (Mason J).

accorded to individuals', corporate bodies may *themselves* receive *direct* protection where *they* experience discrimination 'by reason of the race, colour or national or ethnic origin of any associate of that corporation' (Ibid.). The remaining justices in the majority held for the applicant, without considering it necessary to address the issue of if a corporation may itself sue for discrimination.[15] Although Gibbs CJ, Aickin and Wilson JJ formed the minority (having concluded that the relevant provisions were constitutionally invalid), they also concluded that a corporate body could take the benefit of the race protections as an 'associate' of the individual, asserting that '[p]rovisions . . . which are intended to preserve and maintain freedom from discrimination, should be construed beneficially'.[16] The minority justices also considered that a corporation could claim in its own right under the RDA, on the basis that corporations have a 'national origin'.[17]

Chief Justice Gibbs', Mason's, Aickin's and Wilson JJ's analysis supports the proposition that clause 16 of the RDB extends *indirect* protection to corporate *qua* 'associates' of religious believers. However it must be noted that the limited instances where equivalent 'associate' provisions in the *Disability Discrimination Act 1991* have been interpreted have led to complicated analyses and conflicting results, with Moshinsky J stating that the provisions 'are not free from doubt'.[18] Given the centrality of the associate provision in the protection of religious institutions, and the fact that such provisions are largely untested in anti-discrimination law, we are provided with important reasons to consider alternative means by which a religious corporation can make a complaint of discrimination, whether on the basis of religious beliefs imputed to the organisation or its religious activities (and not solely because of its association with a religious believer). Justice Moshinsky's 'doubt' necessitates such consideration if domestic legislation is to give commensurate effect to the protections afforded under international law.

The judgements in *Koowarta* are not the sole support for this proposition that can be found within Australian law. The Human Rights and Equal Opportunity Commission (the forerunner to the Australian Human Rights Commission) similarly concluded that an Aboriginal community organisation was a 'person aggrieved' for the purposes of the complaint provisions that then existed under the RDA. The HREOC found that the respondents' conduct had prejudicially affected the interests of the organisation because it had hindered it from carrying out its objects.[19] In *Christian Youth Camps Ltd. v Cobaw Community Health Services Ltd.* (*Cobaw*) Redlich JA, in dissent, also recognised that corporate bodies could hold religious beliefs (in the context of exemption provisions), a conclusion that would support the contention that corporations may themselves be subject to discriminatory action on the basis of their beliefs.[20]

However, a different authority takes a contrary position on the question of corporate bodies as complainants in current Australian law. Various Australian courts have held that corporate bodies cannot make complaints under anti-discrimination law. In *IW v City of Perth* Gummow J took the view that an incorporated body of HIV sufferers that had been denied planning approval by a local government could not make a complaint under the *Equal Opportunity Act 1984* (WA), essentially because a corporate body could not contract HIV, and therefore was not capable of an 'impairment',[21] and could not therefore suffer detriment with regard to the protected attribute. Similarly in *Access for All Alliance (Hervey Bay) Inc v Hervey Bay Council* a disability community group's challenge to public transport infrastructure failed because the corporate body could not establish that it was, or could be, 'aggrieved':

---

15    See Ibid., 221 (Stephen J), 242 (Murphy J), 268 (Brennan J).

16    See Ibid., 182 (Gibbs CJ), 243 (Aickin J), 243 (Wilson J).

17    See footnote 16 above.

18    See *Eisele v Commonwealth of Australia* [2018] FCA 15 (24 January 2018) [90] (Moshinsky J). See also *Robinson v Commissioner of Police, New South Wales Police Force* [2013] FCAFC 64.

19    See *Woomera Aboriginal Corporation v Edwards* [1993] HREOCA 24 (extract at (1994) EOC 92-653).

20    *Cobaw* (n 3).

21    See (1997) 191 CLR 1.

> Notwithstanding its intellectual and emotional concern in the subject matter of the proceedings, the interest of the applicant is no more than that of an ordinary member of the public; the applicant is not affected to an extent greater than an ordinary member of the public, nor would the applicant gain an advantage if successful nor suffer a disadvantage if unsuccessful.[22]

In considering if an exemption from discrimination prohibitions for a 'person' could apply to a corporation (as opposed to the distinct question of if a corporation is entitled to religious discrimination protections), Neave JA (in *Cobaw*) determined that a corporate body could not itself hold religious belief:

> Like other human rights, the right to freedom of religious belief can only be enjoyed by natural persons. Because a corporation is not a natural person and has 'neither soul nor body', it cannot have a conscious state of mind amounting to a religious belief or principle. The state of mind of a corporation being a legal fiction, it would be necessary—for the provision to operate intelligibly—for the Court to identify a rule of attribution for the purposes of s 77. This would only be justified if the express provisions of the statutory scheme required for their effective operation the attribution to a corporation of a particular state of mind—in this case, the holding of genuine religious beliefs or principles.[23]

These decisions illustrate the fundamental principle of discrimination law that protections only flow to those who possess an 'attribute'. Appeal Judge Neave's reasoning illustrates the problem: if a religious body cannot hold a belief, it cannot complain of religious discrimination. A religious body cannot assert that discriminatory treatment was 'on the ground of' the body's religious belief if it cannot hold such a belief. Neave JA's view also informs the proposition that a specific provision that outlines the means by which religious belief may be attributed to a corporate body is required within any Federal protection for religious corporations. Part IV considers this further.

Despite the judicial conflict, there are sound policy reasons why religious corporations should be protected. Given religious belief is commonly expressed in associational form (distinguishing it from most of the other protected attributes), any Commonwealth religious discrimination protection and any protection under the AHRA should, in seeking to ensure coverage that encompasses the most common forms of religious activity, should clarify that it extends to the full range of unique religious associational expressions, including both commercial and not-for-profit (including trusts, unincorporated associations, associations, letters patent, corporations, etc.). To fail to do so would allow, for example, that a sole trader taking the benefit of the protections would lose those protections when they subsequently incorporated the business. This would be an arbitrary outcome, as acknowledged by Redlich JA in *Cobaw*:

> That interpretation would produce the unintended result that individuals who operate a business would have different levels of religious freedom, depending upon whether the business was incorporated or not. It would force individuals of faith to choose between forfeiting the benefits of incorporation or abandoning the precepts of their religion.[24]

As discussed further in Part IV, granting religious corporations the ability to access religious discrimination protections does not, of itself, entail third party harm. Further, it is arguable that there may be no particular mischief in not providing a similar right to corporate bodies for other protected attributes, given these attributes do not create the conditions that drive persons to aggregate together in the same way as religious belief. If this was however a concern, commensurate amendments to existing Commonwealth

---

[22]   See (2007) 162 FCR 313, 334.
[23]   See *Cobaw* (n 3) [317] (Neave JA).
[24]   *Cobaw* (n 3) [491] (Redlich JA).

discrimination statutes could be readily made at the time of the introduction of the religious discrimination protections or the AHRA.

### 3.2. Representative Complainant Capacity

Given the differing views, precise articulation of the means by which religious persons may be protected against discrimination is required when this discrimination oc against a religious corporation or group occurs. This section suggests why the RDB and the AHRC's AHRA proposal fail to offer a satisfactory framework. The AHRC submission on the RDB states:

> The Commission considers that a religious body, such as a church, should be able to make a complaint on behalf of its members who have been subjected to discrimination on the basis of their religious beliefs or activities. If it were necessary for such a complaint to go to court, this could be done by way of a representative proceeding.[25]

The AHRC takes a similar position with respect to its proposal for an AHRA:

> [it is recommended that] standing under the Human Rights Act be afforded to individuals who claim that their human rights were breached by public authorities, and organisations or entities acting in the interest of a person, group or class affected by human rights breaches (representative standing).[26]

However, a group's ability to make a direct complaint is separate from the ability to make representative complaints, and it is important to recognise that the two mechanisms address different policy objectives and furthermore, that each mechanism fails to offer a sufficient means to address the bespoke policy concern that the other meets. While representative complainant provisions were, in conjunction with the *Australian Human Rights Commission Act 1986* (Cth) (AHRCA), made available under the RDB, they would not have served the end of protecting businesses, religious institutions or faith-based charities that are *themselves* subject to discrimination through denials of services, funding, contracts and the like. The same is true of corporations asserting a breach of rights under the AHRA (which proposes a remodelled version of the complaint mechanisms in the AHRCA). There are four main reasons for this.

First, as outlined by Dawson and Gaudron JJ in *IW v City of Perth*, in the specific context of a denial of services, it is the person denied the supply who must make the complaint.[27] In emphasizing this, Dawson and Gaudron JJ held that because the complainant was a member of the organisation, as opposed to the corporate body that had been denied planning approval, he was not the person who was refused the supply, and was not therefore subject to discrimination:

> It is clear from the structure of the Act generally and, also, from the structure of Pt IVA, that an 'aggrieved person' is a person who is discriminated against in a manner which the Act renders unlawful. And when regard is had to the precise terms of s 66K(1), it is clear that the person discriminated against is the person who is refused services, or who is provided with services on terms or conditions or in a manner that is discriminatory. As already indicated, there was no refusal of services in this case. And if anyone was the recipient of treatment which might constitute discrimination, it was PLWA, not the appellant. Accordingly, the appellant was not an 'aggrieved person' within the meaning of that expression in s 66A(1) of the Act. And that being so, he is in no position to assert that the City of Perth engaged in unlawful discrimination in the exercise of its discretion to grant or withhold planning approval for PLWA's drop-in centre. (Ibid.)

---

25  See footnote 13 above.

26  See Australian Human Rights Commission, *Free & Equal, Position Paper: A Human Rights Act for Australia,* (2022) 254.

27  See *IW v City of Perth* (1997) 191 CLR 1, 25 (Dawson and Gaudron JJ).

Statutory provisions enabling representative complainant proceedings are thus inapplicable where the corporate body is the body denied services. Such provisions are inadequate to protect religious bodies against denials of funding, tender contracts, or accreditations and the like.

Second, section 46PB(1)(a) of the AHRCA provides that a representative complaint may only be lodged when the class of members are themselves each able to lodge a complaint. The separate means to lodge a complaint 'on behalf' of a person contained in sections 46P(2)(a)(ii) and 46P(2)(c) only becomes available when the person on whose behalf the complaints are lodged is an 'aggrieved person'. The AHRC assertion that these provisions would be available to a corporate body in the absence of a provision clarifying that corporate bodies may make a complaint is, with respect, incorrect. Conceptually, representative complainant provisions have been considered to enable parents to make complaints on behalf of minors, or to enable a person to make complaints on behalf of another person suffering a disability who has suffered discrimination. They do not however enable a person to lodge a complaint on behalf of a corporate body if the corporate body is not itself able to make the complaint. The examples of the corporate entity refused a venue booking or the hospital compulsorily acquired on the ground of its religious beliefs or activities provided above sufficiently illustrate the legal lacuna.

Third, the representative complainant provisions in the AHRCA are styled to enable representation in class-action type of proceedings, not the denial of a specific approval or funding to a corporate body. As Maxwell P, in referring to the Victorian *Equal Opportunity Act 1995* provisions in *Cobaw*, outlined:

> There are several conditions to be satisfied before a complaint may be made by a representative body on behalf of named persons. In particular, each named person must have been entitled, as an individual, to make a complaint of discrimination in his or her own right.[28]

When the discriminatory act is against a corporate business, these conditions will often not exist.

Finally, as Rees, Rice and Allen acknowledge, representative complainant provisions are rarely utilised:

> The legislative provisions governing representative complaints have rarely been used but it is difficult to determine the precise reason for this. Some of the possible reasons include: the provisions are complex; and the remedies available in a representative complaint are usually more limited than those available in an individual complaint.[29]

The scope of protections enabled by such provisions is thus uncertain.

### 3.3. Does International and Foreign Domestic Law Recognise the Rights of Religious Corporations?

Having set out the domestic context and established why additional protection is required beyond what is offered in the RDB and the AHRC proposal for an AHRA, the following section canvasses relevant international law that supports the contention that religious corporations should be protected from discrimination as a means of adequately giving effect to individuals' rights of religious freedom. The AHRC asserts that 'it is an axiomatic principle of international law that human rights extend only to humans'.[30] In itself, this is a non-contentious statement that is consistent with general human rights principles (with the exception of Article 1 of the ICCPR concerning the collective rights of 'peoples'). However, to extend this principle to the absolute conclusion that human rights law precludes corporations from making complaints when discriminatory action has been

---

[28]  See *Cobaw* (n 3) [32] (Maxwell P).
[29]  See (Rees et al. 2018) [15.2.22].
[30]  See Australian Human Rights Commission, *Submission on the Religious Discrimination Bill*: Religious Freedom Bills | Australian Human Rights Commission.

taken against them goes too far, and places a limitation on individuals' rights of religious freedom. As illustrated by the following discussion, a range of leading international bodies and the domestic courts of certain countries have recognised that, due to the unique communal aspects of religious belief, corporate bodies may, and should be able to, assert rights on the basis of their religious beliefs.

### 3.4. United States

As Rienzi notes, in the United States 'both legally and socially, businesses are understood to be capable of having a religious identity if that identity is relevant to their status as a victim of discrimination' (Rienzi 2013). For example, in *Sherwin Manor Nursing Ctr, Inc v McAuliffe* the Seventh Circuit Court of Appeals upheld a complaint of religious discrimination by a privately operated (non-charitable) nursing facility owned and operated by Jews, recognising 'Sherwin presents a cognizable equal protection claim since it alleges that it was subjected to differential treatment by the state surveyors based upon the surveyors' anti-Semitic animus.'[31] Similarly, in *The Amber Pyramid, Inc v Buffington Harbor Riverboats* it was held that a 'minority-owned corporation, like Amber Pyramid, assumes an "imputed racial identity" from its shareholders'.[32] In a 2014 decision (*Burwell, Secretary of Health and Human Services et al. v Hobby Lobby Stores Inc et al* of '*Hobby Lobby*'), the United States Supreme Court held that 'closely held' business corporations can assert religious freedom rights, while acknowledging that '[f]urthering their religious freedom also "furthers individual religious freedom"'.[33] The Court recognised:

> A corporation is simply a form of organization used by human beings to achieve desired ends. An established body of law specifies the rights and obligations of the people (including shareholders, officers, and employees) who are associated with a corporation in one way or another. When rights ... are extended to corporations, the purpose is to protect the rights of these people.[34]

### 3.5. European Court of Human Rights

Although Australia is not bound by the *European Convention for the Protection of Human Rights and Fundamental Freedoms* (ECHR), the European jurisprudence is often adverted to, in acknowledgement of what the Full Federal Court has termed its 'more nuanced and analytical account' of human rights principles.[35] The European Court of Human Rights (ECtHR) has given extensive consideration to the role of corporate bodies in the protection of the right to religious freedom, and therefore offers a specific example that can be considered.

As Ahdar and Leigh recognise, bodies exercising supervisory functions under the ECHR, such as the former European Commission of Human Rights (the Commission) have 'accepted that it was artificial to distinguish between rights of the individual members and of the religious body itself' (Ahdar and Leigh 2013). Accordingly, '[t]he importance of the collective dimension to religious freedom has emerged as an important theme in Convention jurisprudence' (ibid.). In *X and Church of Scientology v Sweden* the European Commission recognised that a church could exercise Article 9 (titled 'Freedom of thought, conscience and religion') rights on behalf of its members:

---

31 See 37 F.3d 1216, 1221 (7th Cir. 1994).

32 See L.L.C., 129 F. App'x. 292, 295 (7th Cir. 2005), (quoting *Thinket Ink Info Res, Inc v Sun Microsystems, Inc*, 368 F.3d 1053, 1059 (9th Cir. 2004)).

33 *Burwell, Secretary of Health and Human Services et al v Hobby Lobby Stores Inc et al, 573 U.S. (10th Cir, 2014)* ('*Burwell, Secretary of Health and Human Services et al v Hobby Lobby Stores Inc et al, 573 U.S. (10th Cir, 2014)*'). There has been some criticism of this decision which is discussed further in Part IV.

34 See *Burwell v Hobby Lobby Stores Inc.*, 134 S Ct 2751, 2768 (Alito J for Roberts CJ, Scalia, Kennedy, Thomas and Alito JJ) (2014) (emphasis in original).

35 *Iliafi v The Church of Jesus Christ of Latter-Day Saints Australia* [2014] FCAFC 26 (19 March 2014) ('*Iliafi*') [70]. See also, for e.g., *Cobaw* (n 3) [322] (Maxwell P), [411] (Neave JA), [481] (Redlich JA).

When a church body lodges an application under the Convention, it does so in reality, on behalf of its members. It should therefore be accepted that a church body is capable of possessing and exercising the rights contained in Article 9 (1) *in its own capacity as a representative of its members.*[36]

The judgement confirms that a church is within the realm of Article 9 rights when acting as a representative of its members, thus providing it with standing to initiate proceedings by relying on Article 9 rights. Citing this authority, the Commission, in *Kontackt-Information-Therapie and Hagen v Austria*, recognised that 'freedom of religion ... can ... be exercised as such by a church'.[37] An array of decisions interpreting the ECHR rights have confirmed that religious corporations may exercise the rights conferred under Article 9 on behalf of their adherents,[38] and this principle is recognised in Australian law. In *Cobaw* Maxwell P regarded this position as 'entirely cogent' and 'properly reflective of the unique function of "church bodies" as institutions in which, and through which, individuals exercise their freedom of religion'.[39]

The ECtHR has also recognised that a corporate vehicle, when acting in this capacity, may legitimately be regarded as the 'victim' of the alleged breach of Covenant rights:

a complaint lodged by a church or a religious organisation alleging a violation of the collective aspect of its adherents' freedom of religion is compatible *ratione personae* with the Convention, and the church or organisation may claim to be the "victim" of that violation within the meaning of Article 34 of the Convention.[40]

Precisely how a corporation may itself be the 'victim' of a violation of the Covenant '*in its own capacity* as a representative of its members'[41] was illustrated by a Court judgement in *Cha'are Shalom Ve Tsedek v France*.[42] Here, the ECtHR recognised the ability of corporations to make religious discrimination claims, taking the benefit of the Article 14 ('Prohibition of discrimination') protections. In this matter the applicant, a Jewish association, considered that the meat slaughtered by an existing Jewish organisation no longer conformed to the strict precepts associated with kosher meat, and sought authorisation from the state to conduct its own ritual slaughters. The organisation's accreditation request was refused on the basis that it was not sufficiently representative of the French Jewish community, and the existence of authorised ritual slaughterers. Notice here that it was the association itself that was the subject of the violation (of Article 9) and the allegedly discriminatory denial (under Article 14). The matter illustrates how it can be said that the corporate vehicle is itself a 'victim', as the Court confirmed in referring to Article 9:

It follows that the applicant association can rely on Article 9 of the Convention with regard to the French authorities' refusal to approve it, since ritual slaughter must be considered to be covered by a right guaranteed by the Convention, namely the right to manifest one's religion in observance, within the meaning of Article 9.[43]

Having determined that 'the facts of the present case fall within the ambit of Article 9 of the Convention', the Court considered 'that therefore Article 14 is applicable'.[44] The Court ultimately concluded, on the basis of a lack of evidence that the refusal would limit

---

36  See *X and Church of Scientology v Sweden* (1979) 16 DR 68, 70 [2] (emphasis added).

37  See *Kontackt-Information-Therapie and Hagen v Austria* No. 11921/86, 57 DR 81 (December 1988).

38  See in particular *Cha'are Shalom Ve Tsedek v France* [GC], No. 27417/95, 27 June 2000; *Leela Förderkreis e.V. and Others v. Germany*, 2008, § 79); *Kontackt-Information-Therapie and Hagen v Austria* No. 11921/86, 57 DR 81 (December 1988), 88; *A.R.m. Chappell v UK*, No. 12587/86, 53 DR 241 (December 1987), 246; *Iglesia Bautisti 'El Salvador' and Ortega Moratilla v Spain* No. 17522/90 72 DR 256 (December 1992).

39  *Cobaw* (n 3) [322] (Maxwell P).

40  See European Court of Human Rights, *Guide on Article 9 of the European Convention on Human Rights Freedom of thought, conscience and religion* (31 August 2022) [11].

41  See footnote 36 above.

42  See *Cha'are Shalom Ve Tsedek v France* [GC], No. 27417/95, 27 June 2000.

43  See Ibid., [74].

44  See Ibid., [87].

the availability of kosher meat to the association's adherents, that the refusal did not comprise discriminatory treatment under the Covenant. Regardless, the matter demonstrates how a corporate entity might *itself* claim it is the 'victim' of a violation of the Covenant. Although the ECtHR found in the circumstances that the organization had not suffered actual disadvantage since it was still able to obtain meat slaughtered by the required method from other sources, it held that the association could itself assert rights under Article 14. The finding that a corporation may itself be the 'victim' of a violation of Article 9 is consistent with the distinction drawn between 'freedom of religion' and 'freedom of conscience' in *Kontackt-Information-Therapie and Hagen v Austria*.[45] In reflecting on this matter, the Commission stated:

> As a private association, the first applicant is a "non-governmental organisation" . . . However, the association does not claim to be a victim of a violation of its own Convention rights. Moreover, the rights primarily invoked, i.e., the right to freedom of conscience under Article 9 (Art. 9) of the Convention . . . are by their very nature not susceptible of being exercised by a legal person such as a private association. Insofar as Article 9 (Art. 9) is concerned, the Commission considers that a distinction must be made in this respect between the freedom of conscience and the freedom of religion, which can also be exercised by a church as such (cf. X and Church of Scientology v. Sweden). The Commission concludes that the first applicant would be debarred from bringing an application invoking Article . . . 9 of the Convention in its own name.[46]

Accordingly, a religious association is not 'debarred from bringing an application invoking Article . . . 9 of the Convention in its own name'.

With respect to State funding to incorporated bodies, *Verein Gemeinsam Lernen v Austria* also confirms that private schools have a right to non-discriminatory conditions of existence, including equal access to State funding for schools of their type.[47] Although concluding that a violation had not occurred in the specific case, the Commission affirmed that Article 14 'requires that any subsidies which are made [to private schools] should not be made in a discriminatory fashion' (Ibid.). That recognition was based upon Article 14 of the ECHR in the context of Article 2 of the First Protocol to the ECHR, whose respective equivalents are Articles 2 and 26 of the ICCPR, and sub-article 18(4). These decisions are consistent with, and give substantive effect to, the principles articulated by the ECtHR in *Hasan & Chuash v Bulgaria* where it stated 'religious communities traditionally and universally exist in the form of organised structures', necessitating a recognition that 'participation in the life of [such communities] is a manifestation of one's religion'.[48] Similarly the Court has recognised that '[w]ere the organisational life of the community not protected by Article 9 of the Convention, all other aspects of the individual's freedom of religion would become vulnerable'.[49]

Rajanayagram and Evans have argued that the 'European case law strikes a sensible balance between the need to allow some entities to bring a case on behalf of groups of individuals, while limiting the capacity of for-profit corporations to do so' (Rajanayagam and Evans 2015, p. 349). They cite the ECtHR's brief statement in *Company X v Switzerland* that 'a limited liability company given the fact that it concerns a profit-making corporate body, can neither enjoy nor rely on the rights referred to in Article 9'.[50] In *Cobaw*, Maxwell P, with whom Neave JA agreed, cited *Company X v Switzerland* alongside his conclusion, which was in accordance with the ordinary rules of statutory construction, that an exemption

---

[45] See No. 11921/86, 57 DR 81 (December 1988) A.

[46] See Ibid., (citations omitted).

[47] See (1995) 20 EHRR CD 78.

[48] See (2002) 34(6) EHRR 1339 [62].

[49] *Fernández Martínez v Spain* (2014) European Court of Human Rights, Grand Chamber, no 56030/07, [127] ('*Fernández Martínez v Spain*').

[50] Application 7865/77 (1981) 16 DR 86, 87. Further applied in *Kustannus OY Vapaa Ajattelija AB and Others v. Finland*, no. 20471/92, Commission decision of 15 April 1996, DR 85.

for 'persons' was inapplicable to religiously affiliated commercial corporate bodies (as he considered the appellant to be), while referencing the fact that the exemption in question sat alongside another exemption for religious corporate bodies. However, as will be argued in Part IV, the question of if an exemption applies to religiously affiliated commercial bodies should be distinguished from the separate question of if such bodies should themselves be protected from discrimination or human rights infringements. *Company X v Switzerland* notwithstanding, the ECtHR has recently appeared to invite further consideration of the question of the protections to be afforded to religiously affiliated businesses. The ECtHR recently said that 'the answer to the question whether an activity which is wholly or partly based on a belief or a philosophy but which is entirely profit-making is eligible for protection under Article 9 is not yet completely clear'.[51] Citing 'recent cases', it also stated that 'the Commission and the Court would appear to leave it open whether Article 9 applies to a profit-making activity conducted by a religious organisation'.[52] In any case, it is clear that Article 9 at least provides a non-profit religious association with the ability to bring an action in its own right. The ECtHR jurisprudence admits that rights inhere in such a corporation in its own capacity when it is itself discriminated against. This principle addresses the issue identified above that relates to representative complaint mechanisms under the AHRCA, and also uncertainties connected with the use of 'associate' provisions, which is similarly identified above.

*3.6. United Nations Jurisprudence*

Article 18(1) of the ICCPR, in express terms, protects the right to exercise the 'freedom, either individually *or in community with others* and in public or private, to manifest his religion or belief in worship, observance, practice and teaching' (emphasis added). General Comment 22 further elaborates:

> The right to freedom of thought, conscience and religion (which includes the freedom to hold beliefs) in article 18.1 is far-reaching and profound; it encompasses freedom of thought on all matters, personal conviction and the commitment to religion or belief, whether manifested individually or in *community with others.*[53]

As Evans notes, 'while human rights belong to individuals, the right to manifest religious freedom collectively means that it has an organisational dimension', whereby it 'is for the individual, rather than the state, to decide whether to exercise the right individually and/or collectively' (Evans 2012).

Article 6 of the *Declaration on the Elimination of All Forms of Intolerance and of Discrimination Based on Religion or Belief* (1981 Declaration), proclaimed by the General Assembly, recognises a range of rights that are by their nature necessarily expressed through corporate vehicles,[54] which include the right 'to establish and maintain appropriate charitable or humanitarian institutions', the maintenance of places of worship, and the observance of ceremonies and holidays.[55] The Declaration has been utilised by the United Nations Human Rights Committee when interpreting the scope of Article 18's protections. In 2005, the UNHRC found that Sri Lanka had breached both Articles 18 (freedom of religion) and 26 (freedom from discrimination) by refusing the incorporation of an order of Catholic nuns whose activities included providing 'assistance to others' as a 'manifestation of religion and free expression'.[56] The complaint was brought by 80 individual sisters, reflecting

---

51  See European Court of Human Rights (n 45) [23].

52  See Ibid., [24].

53  *Human Rights Committee, General comment No. 22 (48) (art. 18),* 48th sess, UN Doc CCPR/C/21/Rev.1/Add.4 (27 September 1993), [1] (emphasis added) (*'Human Rights Committee, General comment No. 22 (48) (art. 18)'*).

54  *Sister Immaculate Joseph and 80 Teaching Sisters of the Holy Cross of the Third Order of Saint Francis in Menzingen of Sri Lanka v Sri Lanka*, Communication No. 1249/2004, U.N. Doc. CCPR/C/85/D/1249/2004 (2005) [7.2].

55  UN General Assembly, *Declaration on the Elimination of All Forms of Intolerance and of Discrimination Based on Religion or Belief*, 25 November 1981, A/RES/36/55, Article 6.

56  *Sister Immaculate Joseph and 80 Teaching Sisters of the Holy Cross of the Third Order of Saint Francis in Menzingen of Sri Lanka v Sri Lanka*, Communication No. 1249/2004, U.N. Doc. CCPR/C/85/D/1249/2004 (2005). (*'Sister*

the procedures under the First Optional Protocol, which only permit complaints from individuals. The UNHRC concluded:

> As to the claim under article 18, the Committee observes that, for numerous religions, including according to the authors, their own, it is a central tenet to spread knowledge, to propagate their beliefs to others and to provide assistance to others. These aspects are part of an individual's manifestation of religion and free expression, and are thus protected by article 18, paragraph 1, to the extent not appropriately restricted by measures consistent with paragraph 3. The authors have advanced, and the State party has not refuted, that incorporation of the Order would better enable them to realize the objects of their Order, religious as well as secular, including for example the construction of places of worship. Indeed, this was the purpose of the Bill and is reflected in its objects clause. It follows that the Supreme Court's determination of the Bill's unconstitutionality restricted the authors' rights to freedom of religious practice and to freedom of expression . . .[57]

The UNHRC's focus was on the effect of the discriminatory denial of incorporated status on the exercise of the individual rights of the members of the body. While a reductionistic focus on individual rights has been challenged by Aroney and Newman,[58] it was clear in any case that the Committee considered that this particular restriction on establishing a corporate vehicle for the Order, through which members could exercise their religious freedom rights, was a violation of Article 18. Their reasoning is summarised by Aroney (albeit in another context) as follows:

> If it is essentially an individual's right to believe and practice, then the freedom will indirectly protect the beliefs and practices of religious groups and organisations in so far as this is necessary to protect the rights of individuals to manifest and practice their religious beliefs.[59]

In *Waldman v Canada*, the UNHRC held that the differential treatment granted by Ontario to Roman Catholic schools (which were publicly funded) over schools of other religions (which were not) amounted to discrimination against the (Jewish) author and other individuals.[60] The distinction drawn by the State could not be considered to be reasonable and objective, and thus violated Article 26. The UNHCR focused on the effect that the treatment of a corporate body would have on individual rights of religious freedom:

> The issue before the Committee is whether public funding for Roman Catholic schools, but not for schools of the author's religion, which results in him having to meet the full cost of education in a religious school, constitutes a violation of the author's rights under the Covenant.[61]

Again, the principle that human rights are enjoyed by individuals did not prevent a successful claim being brought under the ICCPR in connection with differential (discriminatory) treatment of religious entities providing denominational education. To this extent, the decision recognizes that discriminatory treatment of religious institutions that an individual is affiliated to can detrimentally impact their rights.

It should be noted that the First Optional Protocol to the ICCPR recognizes the competence of the UNHRC to receive and determine complaints from individuals claiming to

---

*Immaculate Joseph and 80 Teaching Sisters of the Holy Cross of the Third Order of Saint Francis in Menzingen of Sri Lanka v Sri Lanka').*

[57] *Sister Immaculate Joseph and 80 Teaching Sisters of the Holy Cross of the Third Order of Saint Francis in Menzingen of Sri Lanka v Sri Lanka,* Communication No 1249/2004, UN Doc CCPR/C/85/D/1249/2004 (2005), [7.2].

[58] Aroney (n 8); (Newman 2011).

[59] Aroney (n 8), 154.

[60] *Waldman v Canada* Case No 694/1996, Views adopted on 3 November 1999, [10.5]–[10.6].

[61] Ibid, [10.2].

be victims of a violation by the respondent State of any ICCPR rights.[62] This limitation of individuals is a procedural stipulation of the First Optional Protocol, and does not confine any rights within the ICCPR that individuals exercise collectively. Because individuals enjoy the applicable rights under Article 18, it is technically correct to state that corporate bodiesare not, as such, direct beneficiaries of human rights. However, the right of individuals includes the right to come together collectively in associations. As the foregoing discussion demonstrates, the United Nations jurisprudence recognises that discriminatory treatment against a corporation may impact this individual right.

*3.7. Additional Collective Rights under the Covenant*

Furthermore, it is to be observed that religious freedom is not the only individual right recognised as incorporating the full enjoyment of collective expression under the Covenant. Various Covenant rights incorporate a collective dimension, reflecting the strong emphasis the ICCPR drafters placed upon collective rights as a means of giving effect to individual rights. As leading jurist Manfred Nowak acknowledges, the collective and associational aspects of religious freedom are further supported by referring to Article 22, which protects the 'right to freedom of association with other people'. Nowak explains that this right includes the collective right of an existing association to represent the common interests of its members (Nowak 1993, pp. 386–89). Freedom of association becomes a nonsense if it cannot be exercised through legally incorporated persons.

The Human Rights Committee has also recognised that the freedom of expression under Article 19 necessitates protections to incorporate 'commercial and community broadcasters' or media.[63] This recognises the fact that the legitimate exercise of certain individual Covenant rights can only be fully enjoyed when protections are recognized as extending to incorporated entities. As noted above, Article 18(4) also recognises the liberties of 'parents' extend to the religious and moral education of their children, while Article 23 recognises the family as 'the natural and fundamental group unit . . . entitled to protection by society and the State'. The Human Rights Committee has recognised that 'the persons designed to be protected [by Article 27] are those who belong to a group and share a common culture, religion and/or language'.[64] Article 1 explicitly recognises the collective rights of 'peoples' (although this is the only right for which the collective, rather than the individual, is the direct beneficiary; however, as noted above, the machinery of the Optional Protocol prevents this right being the subject of a complaint to the UNHRC). In addition, although the ICCPR is the primary instrument that the RDB's authority depended on (relying on the external affairs power), it also listed the Convention on the Rights of the Child as an instrument to which it 'gives effect' in Clause 64. Aroney and Parkinson note that Articles 3.2 and 5 concerning the 'responsibilities, rights and duties of parents' and 'the members of the extended family or community as provided for by local custom' 'reflect an understanding that individual rights are often exercised within a social context' (Aroney and Parkinson 2019, p. 9).

**4. Application of the Principles Underpinning International Law**

The underlying principle within both United Nations and ECHR jurisprudence is that measures taken against corporate entities can impact religious freedom or other individual human rights. Further, it recognises that the full enjoyment of rights of religious freedom (alongside other rights) requires the ability to join in mutual action with other individuals. To this extent, ICCPR and ECHR jurisprudence recognises both the individual and collective

---

62 Optional Protocol to the International Covenant on Civil and Political Rights, adopted and opened for signature, ratification and accession by General Assembly resolution 2200A (XXI) of 16 December 1966 entry into force 23 March 1976, Article 1.

63 United Nations Human Rights Committee, *CCPR General Comment No 34 Article 19: Freedoms of opinion and expression*, UN Doc CCPR/C/GC/34 (12 September 2011) [39].

64 See United Nations Human Rights Committee, *CCPR General Comment No 23: Article 27 (Rights of Minorities)*, UN Doc CCPR/C/21/Rev.1/Add.5 (26 April 1994) [5.1], [5.2].

dimensions of religious manifestation and discrimination. Given the propensity of religious belief to inspire collective effort, a failure to provide accordant recognition of, and safeguard from discrimination the vehicles through which this effort is extended would provide incomplete protection. As recognised by the ECtHR, the principle that human rights are enjoyed by individuals does not preclude the ability of a corporate body to initiate a religious manifestation or religious discrimination complaint as a litigant under the European Convention, and this is because of the impact upon the religious exercise of its members.

To provide a concrete and pertinent example, when a government limits a religious institution or school from expressing its traditional view of marriage, this imposes a limitation on the members of the institution effectively exercising their rights through their designated representatives. It also limits the ability of the members to define the particular religious character and ethos of the institution that they have chosen to create, which is what Aroney and Parkinson term as 'the right to shape the identity and culture' of their religious institution.[65] This is the kind of limitation that would enliven a claim to the UNHRC that the restrictions are not authorised under Article 18, and would also enliven a claim under Article 26. Accordingly, the Commonwealth, in operating through religious discrimination legislation, should enable a corporate body to seek a determination within a domestic court as to if direct or indirect discrimination had occurred as a means of recognition, and give substantive effect to protections acknowledged under both Article 18 and Article 26. As Aroney and Parkinson reason:

> if legislative approaches to discrimination policy are to be consistent with the full range of human rights that ought to be recognised and protected, then they should equally recognise and respect the communal aspects of the international human rights standards and their associated jurisprudence.[66]

This is not to say that provision of the capacity for corporations to initiate private litigation is *mandated* by the Covenant, as Article 2(3) imposes an obligation on State parties to offer 'remedies'. The contention is rather that, in applying this framework, there is a strong argument that the Commonwealth *should* provide corporate religious bodies with the ability to make a complaint *qua* a 'person' the subject of a discrimination action, on the basis that this is the most effective method for the Commonwealth to implement its relevant treaty obligations. In other words, since religious individuals have rights under Articles 18 and 22 to create corporations that more effectively exercise their individual rights, and in recognising that it may be necessary for this corporation to take legal measures to preserve the enjoyment of those rights by its members, a State should empower a religious corporation as a litigant, on the basis that, under Article 2, this is the best means of providing a remedy for potential breaches of Covenant rights. Given religious individuals receive protection for religious freedom (including freedom from religious discrimination) from international law through their ability to associate in corporate structures, it would seem necessary to provide such corporate structures with the ability to defend and initiate private actions, in order to enable the protection of individual members to be fully realised.

The 1981 Religion Declaration also states that the right to freedom of religion includes freedom to worship and assemble, establish charitable and humanitarian institutions, and appoint appropriate leaders, who are consistent with the requirements and standards of the religion.[67] It follows that there is a close connection between Article 18 of the ICCPR and other fundamental human rights, including freedom of association (Article 22 of the ICCPR). Under the ICCPR, freedom of religion in conjunction with freedom of association, thus protects the right to found an association based on a common purpose, along with

---

[65] See Ibid, 12–13.

[66] Ibid, 19–20. See also (Ahdar 2016).

[67] (Aroney 2019, pp. 711–12); *Declaration on the Elimination of All Forms of Intolerance and Discrimination based on Religion or Belief*, GA Res 36/55, UN GAOR, 36th sess, 73rd plen mtg, Supp No 51, UN Doc A/RES/26/55 (25 November 1981) Art 6.

the right of this association to be recognised as and function as a distinct legal person, and to select and regulate members in accordance with the common interest of the association (including expulsion of those who breach the terms of the association.[68] Further relevant obligations of States that have ratified the ICCPR include the 'right of a group to a legal framework making possible the creation of juridical persons' and 'the collective right of an existing association to represent the common interests of its members'. These two rights necessarily entail the ability of religious corporations to sue in their own right, including in relation to discrimination claims.[69] 'Religious communities need to be able to secure legal personality status within a society in order to exercise many of their collective religious freedoms'.[70] Articles 22 and 27 of the ICCPR also protect the right of freedom of association in community with others. As noted above, Article 6 of the 1981 Declaration concordantly affirms an array of freedoms that are communal in expression and necessitate the recognition of legal personality, such as the maintenance of places of worship and the establishment of charitable institutions. The overlapping protections of the ICCPR and the 1981 Declaration suggest that, although they are independent stand-alone rights under international law, there are certain circumstances when freedom of religion requires freedom from religious discrimination. And if the full enjoyment of these rights are to be realised, domestic law should recognise the capacity of religious corporations to be litigants.[71] As Aroney concludes:

> . . . since international human rights law recognises that religious freedom extends to the establishment and maintenance of religious, charitable, humanitarian and educational institutions, and the right to establish associations with like-minded people includes the right to determine conditions of membership and participation within such organisations, consideration should be given to protecting freedom of religion in the context of anti-discrimination laws [which includes the ability of such bodies to make discrimination claims].[72]

As the developed jurisprudence under the European Convention on Human Rights also demonstrates, the protection of religious corporations from discrimination is entirely consistent with human rights principles, and is an effective mechanism by which to guarantee the adequate protections of individual human rights.

*Attributing 'Sincere' Religious Beliefs to Corporations*

With regard to the functional question of how religious belief may be meaningfully attributed to a corporate body, the *Hobby Lobby* case mentioned earlier offers some pertinent insights. It concerned a private company that wanted a religious exemption under the US *Religious Freedom Restoration Act* ('RFRA'); the desired exemption was from obligations under the healthcare mandate, because business owners did not want to be morally complicit in abortions by funding reproductive healthcare insurance for their employees.[73] The Supreme Court held (by 5:4) that this exemption should be granted because a private company can exercise the right to religious freedom and belief.[74] The Supreme Court held that a corporation is a legal person that can raise religious freedom claims under the RFRA.[75] Although corporations may pay the penalty, 'the humans who own and control these companies' are subject to the burden imposed on religion.[76] Under RFRA there is a requirement to prove a substantial burden on religious exercise for an exemption to be granted, and a major issue that arose in *Hobby Lobby* is how a corporation proves substantial

---

[68]   See Aroney (n 77) 712; Nowak (n 71) 386–389; (Rivers 2010, pp. 34–38).
[69]   See Aroney and Parkinson (n 74) 8.
[70]   See Ibid., 10.
[71]   Ibid., 11; See the discussion in (Taylor 2005, pp. 235–92).
[72]   See Aroney (n 77) 720.
[73]   See *Burwell v Hobby Lobby Stores, Inc.* 134 S. Ct. 2751 (2014) ('Hobby Lobby').
[74]   For a detailed analysis from various perspectives, see (Schwartzman et al. 2016).
[75]   See *Hobby Lobby* (n 83) 2768–70.
[76]   See Ibid., 2768.

burden. As Hardee explores, there are difficulties with determining the religious sincerity of a corporation when its members or shareholders may have diverse religious convictions. She recommends a dual inquiry into the veracity of the shareholders' religious beliefs and an attribution inquiry that determines whose religious beliefs should be considered when determining the sincerity of the corporation.[77]

In *Hobby Lobby* the Supreme Court indicated that they would be 'deferential to a religious institution's claim of a burden on free exercise', and established the Court's 'narrow function... is to determine whether the plaintiffs' asserted religious belief reflects an honest conviction' (Alvare 2019, p. 158). The Court rejected the argument because the complicity was too attenuated to be cognizable, meaning it could not engage in a theological analysis of if the burden was true and substantial, and accordingly held 'a sincerely held belief that results in severe penalties is sufficient to establish a substantial burden under RFRA' (Carmella 2020, pp. 581–82). Hence, an appropriate solution for evidencing the religious belief of a corporation is to accept the testimony of the religious corporation as to its sincerity, along with the content and nature of its belief, as expressed by those within the corporation who possess the recognised authority to speak on behalf of the corporation.[78] As Fowler has argued, such reasoning reflects the approach adopted by the New South Wales Court of Appeal in *OV & OW v Members of the Board of the Wesley Mission Council*.[79] Therefore, in any discrimination protection extended to religious corporations, there should be a provision that requires deferral to the religious corporation's articulation of its relevant religious beliefs and/or activities, whose provision necessitates an evidential inquiry into the locus of the authoritative expression of these beliefs by the corporation. The articulation of beliefs and/or the wider activities of the corporation must also provide supporting evidence of the sincerity (though not the truth) of these beliefs. Such an approach would, in substance, apply the tests developed by courts for assessing the sincerity of individual beliefs in the context of incorporated bodies.[80]

Despite this functional solution, Rajanayagam and Evans (in commenting on *Hobby Lobby*) argue that 'corporations should not be understood to have the right to religious freedom'.[81] Their central rationale is that to allow religious corporations 'to put forward religious freedom arguments would be at odds with the separate legal personality doctrine; it would allow the shareholders to pierce the corporate veil in reverse, attributing their sincerely held beliefs to the corporation'.[82] They assert: 'Those who live by the corporate form must, of necessity, sometimes die by it as well'.[83] This rationale notwithstanding, they accept what they characterise as the ECtHR distinction between protected not-for-profit religious institutions and non-protected for-profit vehicles:

> At the extreme end of the scale, for example, if a church were forced to shut down by a hostile government, it would be possible for a large number of individuals to bring a claim that their religious rights had been breached because they are no longer able to worship and practise their religion as they once could. However, it is more efficient for the courts, and beneficial for the impacted individuals, for the church to be able to bring a case itself—even if it has taken the corporate form.[84]

---

77    (Hardee 2020, pp. 1764–65). See also (Fowler 2021; Foster 2020).
78    See (Deagon 2021, pp. 60–87); Foster (n 87).
79    *OV & OW v Members of the Board of the Wesley Mission Council* (2010) 79 NSWLR 606 ('*Wesley Mission*'). See Fowler (n 87).
80    See, for e.g., *Church of the New Faith v Commissioner for Pay-roll Tax (Vic)* (1983) 154 CLR 120, 170 (Wilson and Deane JJ), 141 (Mason ACJ and Brennan J); *R (on the application of Williamson) v Secretary of State for Education and Employment (Williamson)* (2005) 2 AC 246, 267 (Lord Walker); *Syndicat Northcrest v Amselem* (2004) 2 SCR 551; *Employment Division, Department of Human Resources of Oregon v Smith* (1990) 494 US 872 at 886–887 (Scalia J giving the majority opinion).
81    Rajanayagam and Evans (n 56) 332. The *Hobby Lobby* case, in particular, received significant criticism. However, in this article we focus on relevant general legal and human rights principles, rather than on the US decision itself. For critical commentary of the decision see e.g., (Lupu 2015; Greenawalt 2015).
82    See Rajanayagam and Evans (n 56) 342.
83    See Ibid., 356.
84    See Ibid., 349.

As Rajanayagam and Evans acknowledge, '[i]f religious groups were not able to bring cases on behalf of their members, the courts would lose a sensible and valuable vehicle to assist with the protection of individual rights' (Ibid.). As Fowler has written elsewhere, there is an international law consensus on recognising faith-based charities as religious bodies (Fowler 2020). Applying this understanding of human rights law to Rajanayagam's and Evans' analysis would support affording religious discrimination protections to, for example, Calvary Public Hospital. Under ECtHR jurisprudence, this hospital could be viewed and treated as the 'victim' of a violation '*in its own capacity* as a representative of its members'.[85]

> However, Rajanayagam and Evans draw a line at for-profit bodies, claiming that the focus should be on if it is essentially the religious organisation itself that is a party to the case, representing the interests of its members, or instead an entity with religious affiliations (but one whose principal reason for existing is profit maximisation rather than advancing a religious doctrine) that is bringing a claim that should properly be brought by its members.[86]

Precisely how allowing not-for-profit incorporated bodies but excluding for-profit incorporated bodies can sit comfortably with their concern to preserve the United States and Australian 'separate legal personality doctrine' is not clear. Nevertheless, two further objections can be expressed against applying their view to deny protections against religious discrimination to religiously-affiliated corporations.

The first is that the argument fails to take into account, and does not reference, the leading contrary Australian authority, *Commissioner of Taxation v Word Investments ('Word Investments')*. In this matter, a majority of the High Court held that a company conducting purely commercial operations with a purpose of raising a surplus to be gifted to religious organisations 'has only one group of objects–a group of objects of advancing religious charitable purposes'.[87] Rajanayagam and Evans' exclusion of entities whose 'principal reason for existence is profit maximisation, rather than the advancement of religious doctrine', defies the High Court's characterisation. Citing *Word Investments*, Aroney has argued: 'even commercial' organisations 'formed for . . . religiously-oriented purposes' 'are in principle able to enjoy the protections of s 116 [of the Australian Constitution] just as much as any individual', '[p]rovided, it seems, that their establishment or their activities are relevantly an "exercise of religious freedom"'.[88] In also citing the High Court majority in *Word Investments*, and commenting on the conscientious denial of a service in *Cobaw*, Aroney, Harrison and Babie argue

> [f]undamentally, religious liberty has traditionally been understood as protecting against a coerced conscience, and as securing the capacity of groups to pursue an ethos and form a body of people in response to an understanding of the divine. On this view, an act . . . does not necessarily cease to be an act of religious conscience simply because it takes place in a commercial setting. (Aroney et al. 2017)

Rajanayagam's and Evans' claim that 'impositions on the [religiously affiliated commercial] corporation do not directly impact individual rights in the same way'[89] as those on non-profits simply does not withstand this view of religious freedom and conscientious protection. As human rights law holds, the impact upon the individual is relevant when we consider discriminatory treatment of any corporations with which they are affiliated. There are circumstances when a limitation upon a corporation can have the same impact on the conscience of a related individual as when an action is taken directly against the individual. Appeal Judge Redlich's assessment that an identical burden can be imposed on the religious conscience of a sole trader and a director of a corporation is in this sense

85  *X and Church of Scientology v Sweden* (1979) 16 DR 68, 70 [2] (emphasis added).
86  Rajanayagam and Evans (n 56) 349.
87  *Commissioner of Taxation of the Commonwealth v Word Investments Ltd* (2008) 236 CLR 204, [19] ('*Word Investments*').
88  Aroney (n 67) 157–8. See also (Murray 2009).
89  See footnote 86 above.

compelling. As Redlich JA stated, excluding incorporated businesses leads to 'the unintended result that individuals who operate a business would have different levels of religious freedom, depending upon whether the business was incorporated or not'.[90] The line between commercial and religious purposes and activities is therefore not so readily maintained.[91]

This leads to the second objection to Rajanayagam and Evans' distinction: the unique policy considerations that arise in the context of protections for persons against *religious discrimination*. In discrimination law, the central test is not the overall characteristics of the 'person' discriminated against, but rather if the allegedly discriminatory action has been taken 'on the ground of' their religious belief or activity. Each Commonwealth, State and Territory Act operates upon the substantive recognition that it is irrelevant 'whether or not the particular matter is the dominant or substantial reason for the doing of the act',[92] or words to this effect. Rajanayagam and Evans' concern is that '[h]aving conferred upon the shareholders the significant benefits of corporations law, it therefore seems inequitable to allow them to discard the separate legal entity doctrine merely because it suits their interests in a particular legal context'.[93] However, it would seem a greater inequity, as Redlich JA correctly identifies, to refuse to offer protection from religious discrimination because of the corporate form an entity takes (whether they are a for-profit or not-for-profit entity, as Rajanayagam and Evans suggest), notwithstanding the fact that the detrimental action in question could be evidenced to the standard afforded in discrimination law for being the 'ground of' the action. As noted above, the proposition that a burden upon religious conscience is not felt in the same way on account of the corporate form that a religious person is associated with is questionable.

Thus, there is simply no logical basis for asserting that a corporation that is discriminated against on the basis of the religious beliefs of its owners should be excluded from 'the exercise of a right on behalf of the members of the religious group that the corporation represents' simply because it operates a business.[94] And even if this is not accepted, Rajanayagam and Evans have already conceded that such protections can and should at least be afforded to non-profit religious groups. Rajanayagam and Evans are concerned that to admit religious corporations may assert rights of religious freedom means 'in most cases of this kind [that] there is third party harm' enacted upon others by religious persons.[95] However, this problem is not necessarily applicable when the harm has been experienced by individuals on the basis of their religious beliefs. Where third party harm exists, the impact of this harm can be weighed by decision makers determining if discrimination had occurred by using the standard statutory tests (including the 'on the ground of' test). The prospective existence of harm therefore provides no reason to deny statutory protection to religious believers on the grounds they do not assume incorporated forms. The policy rationales cited by Rajanayagam and Evans for not piercing the veil are not infringed upon if religious corporations are permitted to benefit from religious discrimination protections. For these reasons, Rajanayagam's and Evans' argument that 'denying' corporations 'the right to freedom of religion' 'would not unduly burden most religiously motivated individuals' is not accurate. For the reasons outlined in Part III, where the uncertainties of 'associate' mechanisms and the insufficiency of the representative complaint regime were discussed, the claim that denial of the right to corporations would not 'exclude religious bodies from bringing cases on their behalf' is also incorrect.[96]

---

90    See footnote 24 above.
91    For further support on this point, see e.g., (Horwitz 2014, pp. 177–81); Ahdar (n 76) 4.
92    See *Sex Discrimination Act 1984* (Cth), s 8.
93    Rajanayagam and Evans (n 56) 355.
94    See Ibid., 350.
95    See Ibid., 351.
96    See Ibid., 332.

### 5. Constitutional Considerations

Given it has been argued that the Commonwealth should pass a religious discrimination law empowering religious corporations as litigants, this part briefly provides two independent constitutional justifications for such a law. First, the Commonwealth may rely on the 'external affairs' power, which enables the Commonwealth to make laws for the purpose of implementing rights and obligations arising from international treaties ratified by Australia.[97] In the seminal *Victoria v Commonwealth* (*Industrial Relations Act*) case, the High Court outlined the applicable test: the law 'must be reasonably capable of being considered appropriate and adapted to implementing the treaty', and the law 'must prescribe a regime that the treaty has itself defined with sufficient specificity to direct the general course to be taken by the signatory states'.[98] The first aspect (conformity) entails a proportionality analysis that considers the purpose of the treaty and that recognises 'it is for the legislature to choose the means by which it carries into or gives effect to the treaty'.[99] The second aspect (specificity) requires that the treaty embodies precise obligations, rather than mere aspirations, which are 'broad objectives' permitting 'widely divergent policies'.[100]

As explained above, the ICCPR and associated instruments prescribe a clear right to freedom of religion that includes freedom to manifest religion in a community with others. Manifesting religion in community with others entails the creation and continuance of incorporated and unincorporated religious associations that function as distinct legal persons for a common purpose. Article 18 recognises the ability of persons to form and incorporate religious associations as a function of exercising their rights of freedom of religion and association. This right may be given effect by the ability of such bodies to seek redress in the event of illegitimate limitations of religious manifestation. In addition, Articles 2 and 26 recognise an obligation to not discriminate against religious adherents, and this obligation is also given effect by the ability of such bodies to seek redress in the event of such discrimination. The Article 26 right also correspondingly entails the ability of religious individuals to seek redress against a body in the event of discrimination. As such, a Commonwealth law that prohibits religious discrimination is reasonably capable of being considered appropriate and adapted to the purpose of implementing relevant international law obligations. The purpose of such a law may legitimately extend to protecting the religious freedom of religious corporations by protecting them against discrimination in their own right. Though this is not mandated by the Covenant, it is for the legislature to choose the means of implementation. The Commonwealth would be enacting a law that implements obligations in a treaty, or secures benefits under a treaty. Such an implementation of the international framework through domestic legislation asserts the same effective basis in the external affairs power, as is found in existing Commonwealth anti-discrimination Acts that offer private parties recourse against one another and against government within domestic courts. The addition of an ability for religious corporate bodies to stand as parties in private litigation is therefore a form of recourse that the Commonwealth has the power to provide, and is the most effective means to give sonorous and commensurate effect to the relevant international protections.

Second, the Commonwealth can empower religious corporations as litigants by exercising the corporations power. Section 51(xx) of the Constitution enables the Commonwealth to make laws with respect to 'foreign corporations, and trading or financial corporations formed within the limits of the Commonwealth'. These kinds of corporations are known as 'constitutional corporations'. This presents two questions relating to the constitutional validity of the RDB's empowerment of corporations as litigants: first, are religious corpo-

---

97   *Koowarta v Bjelke-Petersen* (1982) 153 CLR 168. See (Saunders 1995, p. 159).

98   See *Industrial Relations Act* (1996) 187 CLR 416, 486-487 (Brennan CJ, Toohey, Gaudron, McHugh and Gummow JJ).

99   See Ibid., 487.

100   Ibid., 486. Though the 'absence of precision does not . . . mean any absence of international obligation.' See *Tasmanian Dams* (1983) 158 CLR 1, 261-2 (Deane J).

rations constitutional corporations? And second, does the corporations power extend to authorising the definition of religious corporations as litigants?[101]

A religious corporation will be a constitutional corporation if it is formed outside the limits of the Commonwealth of Australia under the law of a foreign nation but operates within Australia.[102] Alternatively, a religious corporation is a constitutional corporation if it is a trading or financial corporation that is formed, and operates within, the limits of the Commonwealth. The test is if trading or financial activities form a substantial or significant proportion of the corporation's activities.[103] Aroney and Turnour observe that while such trading or financial activities are usually undertaken for the purpose of earning money, it is not essential for the corporation to be seeking a profit. Charitable and civil society corporations, such as the Australian Red Cross, have been held to be trading corporations,[104] This is a point that is obviously relevant to religious corporations. The majority would not pursue profit as their primary objective, and would instead orientate towards more altruistic and community-oriented activities; however, if a significant proportion of their activities are trading or financial activities, then they too will be regarded as constitutional corporations.

In terms of the extent of the corporations power, the High Court has taken a very broad approach. The Court has upheld legislation that is not only connected to the trading or financial activities of corporations, but also to the trading or financial corporations themselves. It has been observed that: 'The Court has also upheld legislation that prohibits conduct by some other party that is intended or likely to cause loss or damage to a constitutional corporation'.[105] There must be a 'sufficient connection' between the law and the constitutional corporation, and more recent cases have indicated that a sufficient connection is established when a law imposes a duty or confers a right upon a constitutional corporation, singling it out as 'an object of statutory command'.[106] In effect, this means that if a law regulates or instructs constitutional corporations, this law will be supported by the corporations power. The power therefore also extends to the business relationships of constitutional corporations, and to 'persons by and through whom they carry out business function and activities'.[107] As Gaudron J observed (again in dissent, and again endorsed by the majority in *Workchoices*), the power extends to all activities, functions and relationships of constitutional corporations, including the imposition of obligations on corporations and the regulation of those whose conduct affects these aspects of corporations.[108] In particular, the case of *Actors Equity* held that the corporations power supports laws regulating what third parties may do with respect to trading or financial corporations.[109] As noted by Gibbs CJ for the Court, the legislative purpose that was upheld was the protection of corporations: 'the conduct to which the law was directed is conduct designed to cause, and likely to cause,

---

[101]    See (Aroney and Turnour 2017, pp. 475–81) for a similar analysis.

[102]    See *New South Wales v Commonwealth ('Incorporation Case')* (1990) 169 CLR 482.

[103]    *R v Federal Court of Australia; Ex parte The Western Australian National Football League (Inc)* (1979) 143 CLR 190, 233 ('Adamson's Case'). See (Tran 2012).

[104]    See Aroney and Turnour (n 116) 476.

[105]    See Ibid., 477.

[106]    *New South Wales v Commonwealth* (2006) 229 CLR 1, 115–16 [179]–[181], 121 [198] ('*Workchoices*'). C.f. (Blackshield 2007). It is in principle arguable that a law preventing discrimination against constitutional corporations is sufficiently connected to these corporations. See also (Glover 2009).

[107]    *Re Dingjan; Ex parte Wagner* (1995) 183 CLR 323, 365 (Gaudron J). While Justice Gaudron was in dissent, her judgment was subsequently endorsed by the majority in *Workchoices* 114–15 [178]. The implications of this decision are extensive and mean there are potentially very few, if any, corporations that cannot be regulated by the Commonwealth.

[108]    *Re Pacific Coal Pty Ltd.; Ex parte Construction, Forestry, Mining and Energy Union* (2000) 203 CLR 346, 375 [83], quoted in *Workchoices* 114–15 [178]. In addition, the definition of 'entity' in the *Australian Charities and Not-for-profits Commission Act 2012* captures incorporated and unincorporated bodies and the full range of trusts, partnerships, etc, which are prevalent in the not-for-profit sector. The ACNC Act clearly proceeds on the basis of an understanding that Constitutional power exists to regulate these entities. There is no relevant distinction that would preclude the same assumption from applying to the protection of these bodies from religious discrimination.

[109]    See *Actors and Announcers Equity Association of Australia v Fontana Films Pty Ltd.* (1982) 150 CLR 169 ('Actors Equity').

substantial loss or damage to the business of a trading corporation. . . A law may be one with respect to a trading corporation, although it casts obligations upon a person other than a trading corporation'.[110]

Hence, categorising religious constitutional corporations as litigants under a Commonwealth prohibition on religious discrimination can be supported by the corporations power on two related grounds. First, following *Actors Equity*, such a law would protect religious constitutional corporations from loss or damage resulting from discrimination against them, imposing obligations on third parties to not discriminate against them. Second, following *Workchoices* more broadly, such a law would make religious constitutional corporations the object of statutory command by regulating the potentially discriminatory actions of religious constitutional corporations and those who engage with them. Specifically, religious constitutional corporations cannot discriminate and cannot be discriminated against by other persons. This confers a right upon religious constitutional corporations to not be discriminated against and an obligation upon the actions of religious constitutional corporations to not discriminate. Therefore, conferring protection from discrimination on a religious constitutional corporation, and preventing such corporations from discriminating, therefore falls within the core of the corporations power.

## 6. Conclusions

This article responds to the claim that it is contrary to human rights principles and international law to empower religious corporations as litigants in human rights and anti-discrimination law. International law obligations protect the freedom of individuals to manifest their belief in community with others by forming incorporated or unincorporated associations. The protection needs to be combined with a prohibition of discrimination against religious associations, which implies the ability of such associations to be litigants in their own right, by acting as a corporate representative of their members. This principle is supported in both the domestic jurisprudence of nations and in the international law jurisprudence of the United Nations and the European Court of Human Rights. It addresses the issues associated with representative complaint mechanisms and the uncertainties attached to the use of 'associate' provisions. The article has also briefly indicated how such a law may be constitutionally supported. Empowering religious corporations as litigants in both human rights and anti-discrimination law gives full and proper effect to existing international law obligations and jurisprudence pertaining to freedom of religion and association.

**Author Contributions:** Both authors contributed equally to all parts of the manuscript. All authors have read and agreed to the published version of the manuscript.

**Funding:** This research received no external funding.

**Data Availability Statement:** Data are contained within the article.

**Conflicts of Interest:** The authors declare no conflict of interest.

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
