# Peer review of "Recognising Religious Groups as Litigants: An International Law Perspective"

_laws, 2023_

Round 1

Reviewer 1 Report

Comments and Suggestions for Authors

This paper makes a sound and accurate contribution to contemporary Australian debate. It is ironic that the personnel of the Australian Human Rights Commission should misunderstand the nature of religious freedom set out in international instruments so badly that they do not think it extends to associations and corporations of religious believers. But because some of their publications demonstrate that error, this piece is a timely corrective. I have made some editorial comments and one recommendation as to substance towards the conclusion of the article. That matter of substance is minor but should be referred to the author/s for consideration. It is that since the Work Choices Case was decided in 2006, I do not think it accurate to suggest that the Australian Commonwealth cannot regulate corporations which are not either foreign, financial or trading in their primary character. In practice, since that decision was handed down, the Australian Commonwealth has been able to regulate any association howsoever incorporated.

Comments on the Quality of English Language

No concerns at all. Some minor editorial suggestions, but they are not compulsory.

Author Response

The substantive edit requested on page 24 concerning work choices has been made.

Most other minor recommendations have been accepted.

Reviewer 2 Report

Comments and Suggestions for Authors

Author Response

No changes were required, and none were made respecting this specific review.

Round 2

Reviewer 1 Report

Comments and Suggestions for Authors

Nil

Author Response

Three further revisions have been requested. The first two on pages 9 and 10 are formatting suggestions, which have been followed. The final one on page 19 concerns a glitch in the document, which has been remedied.